# Capacity optimization configuration and multi-dimensional value evaluation of integrated energy system with power-to-hydrogen

**Kuiyuan Pan, Tianhe Sun**⊙*, **Xinfu Pang**⊙, **Xiaoyi Qian**

Key Laboratory of Energy Saving and Controlling in Power System of Liaoning Province, Shenyang Institute of Engineering, Shenyang, P R China

* sunth@sie.edu.cn

## Abstract

The research on the value evaluation system of power-to-hydrogen (P2H) equipment configuration in integrated energy systems is of great value for optimizing resource allocation, improving energy utilization efficiency, and promoting clean energy technology development. However, there is no comprehensive evaluation system for evaluating P2H equipment configuration in integrated energy systems. Therefore, a multi-dimensional value evaluation system is proposed to realize the thorough evaluation of P2H equipment with different capacity configurations in the integrated energy system. Initially, a mathematical model considering flexibility benefit, new energy consumption benefit, economic benefit, and environmental benefit is established to maximize the comprehensive benefits brought by P2H equipment to the integrated energy system, and the model is solved using an improved backbone particle swarm optimization (IBBPSO) algorithm; subsequently, a multi-dimensional value evaluation system based on the analytic hierarchy process (AHP) -entropy weight method is constructed, and the value of P2H equipment with different capacity configurations in the integrated energy system is compared and analyzed when the comprehensive benefit is optimal. The experimental results show that the IBBPSO algorithm exhibits better performance in solving the optimization model. Compared to PSO, IBBPSO, GWO, and WOA algorithms, it improves by 9.8%, 11.09%, 33.57%, and 17.7%, respectively. The optimal solution is achieved when the P2H equipment is configured to 50 MW.

## 1 Introduction

The electrolytic hydrogen production technology driven by new energy power is conducive to expanding the scale of hydrogen production from renewable energy in China and achieving the goal of carbon neutralization in advance. This technology will become a competitive choice, strongly supported by government policies [1]. Under the background of the " double carbon " goal to promote the rapid development of new energy, the comprehensive and accurate evaluation of the P2H equipment in the integrated energy system (IES), as well as the

**Data availability statement:** All relevant data are within the manuscript and its Supporting Information files.

**Funding:** This research was funded by the Shenyang Young and Middle-aged Science and Technology Innovation Talent Support Program ( RC220456). The funders had no role in study design, data collection and analysis, decision to publish, or preparation of the manuscript.

**Competing interests:** The authors have declared that no competing interests exist.

rational allocation in different application scenarios, has become one of the key measures to promote its rapid development and optimize resource allocation.

## 1.1 Literature review

Capacity configuration is essential to ensure system resources are efficiently allocated and used to improve performance, reduce costs, and meet specific business needs. Some scholars have combined two energy conversion technologies to establish a hybrid energy system (HES) and optimized its capacity configuration to improve energy efficiency and system reliability. Reference [2] discusses hydrogen storage's potential in HES and establishes an economic evaluation model. In addition, a HES model including a reversible solid oxide battery (RSOC) has been established, which effectively reduces the power mismatch phenomenon [3]. Further, Reference [4] considers the impact of economic uncertainty on HES. Reference [5] establishes a photovoltaic-hydrogen hybrid energy system and analyses economic and environmental benefits and policy issues. To better manage various energy forms and optimize energy efficiency, some researchers have established a multi-energy system (MES). Reference [6] proposes a novel MES optimization model and discusses two hydrogen production methods: gas-to-gas and electric-to-gas. In addition, Reference [7] proposes a PV/battery/hydrogen MES for hydrogen production and establishes a stable optimization model to resist hydrogen uncertainty effectively. IES can provide comprehensive energy solutions, maximize energy efficiency, and promote sustainable development, so capacity configuration research is also essential. Reference [8] establishes an IES capacity configuration optimization model of cold-determined heat and heat-determined electricity. From the perspective of environmental benefits and economic analysis, cold and heat priority strategy can be used to maximize system efficiency and minimize carbon emissions. Reference [9] proposes a robust optimization model for traditional IES systems that considers demand response and thermal comfort. Reference [10] proposes an expansion planning model considering hybrid energy storage, which improves the utilization rate of renewable energy. In addition, an IES combining combined heat and power (CHP) and heat storage tank (HST) is established, and the influence of heat load on the capacity configuration of HST is analyzed [11]. These studies have shown that system capacity configuration significantly promotes intelligent and green transformation of energy systems.

The problem of system capacity configuration is a complex problem with multiple constraints and uncertainties, especially in HES, MES, and IES. To solve this problem, researchers have proposed various methods. Classical optimization algorithms can efficiently solve linear or nonlinear problems that can be transformed into convex problems to obtain accurate global optimal solutions. For example, Reference [12] formulates the optimal planning problem as a mixed integer linear programming (MILP) model to minimize the total annual cost. Reference [13] proposes a solution method based on sequence operation theory (SOT), which converts the scheduling model based on chance-constrained programming (CCP) into a linear programming problem that can be solved. Reference [14] proposes a least squares approximation method that simplifies the complex trigonometric functions representing the conversion efficiency of the bidirectional converter (BDC) as it varies with power, transforming the original non-convex relationship into a computationally efficient convex form. Although classical optimization algorithms have the advantages of high reliability and high-quality feasible solutions for solving linear problems, they cannot deal with highly nonlinear and uncertain problems. In these cases, meta-heuristic algorithms need to be combined to solve them. To improve the sustainability of energy production infrastructure in remote areas, some

researchers have adopted a hybrid meta-heuristic algorithm [15], combining two or more algorithms to take advantage of their respective advantages to improve the solution's quality and the algorithm's convergence speed. Reference [16] proposes a hybrid Runge Kutta-gradient-based optimization algorithm, which combines the Runge Kutta optimizer and a gradient-based optimizer to solve a generation and transmission expansion planning model embedded with energy storage systems. Reference [17] develops a hybrid artificial rabbits sine-cosine algorithm to address complex non-convex problems in AC optimal power flow models. Reference [18] proposes a hybrid optimization algorithm (BAPSO) that combines particle swarm optimization (PSO) and bat algorithm (BA) to optimize solar power generation capacity. Reference [19] combines the features of the Marine Predators Algorithm (MPA) and the Honey Badger Algorithm (HBA) to propose a hybrid algorithm (MPA-HBA) for solving the integrated hosting capacity model for electric vehicles. A study also combines multi-objective particle swarm optimization (MOPSO) and TOPSIS algorithm to solve electric vehicles' charging and discharging model considering demand response [20]. To improve the energy supply effect of the system, Reference [21] proposes a regional optimization design method combining K-means and genetic algorithm (GA). Multiple objectives, such as economic efficiency and environmental benefits, are often considered when configuring system capacity. Therefore, scholars have designed improved multi-objective optimization algorithms to obtain better solutions. Reference [22] introduces a multi-objective capacity configuration model and solves it using an improved hybrid multi-objective particle swarm optimization algorithm (HMOPSO). In another study, to reduce energy costs, reduce the probability of power failure, and increase the proportion of renewable energy use, a multi-objective variant of the crow search algorithm (MOCSA) is designed to solve the problem [23]. Reference [24] proposes an improved multi-objective artificial hummingbird algorithm (MOAHA) to optimize the distribution and size adjustment of distributed generation and battery energy storage system (BESS). In addition, many heuristic and meta-heuristic algorithms are applied to the capacity configuration problem. The improved particle swarm optimization (IPSO) is applied to the optimization of microgrid capacity [25,26], and the cat swarm optimization (CSO) is used to optimize the capacity of the gravity energy storage system [27]. There are also other improved algorithms, such as the improved pelican optimization algorithm (IPOA) [28], improved genetic algorithm (GA) [21], golden eagle algorithm (IGEO) [29], and NSGA-II algorithm [30–32]. These intelligent optimization strategies show higher flexibility and efficiency than traditional methods. The research status of key studies is summarized in Table 1.

## 1.2 Research gap and contributions

By reviewing and analyzing the literature on capacity configuration in the system, most of the system capacity configuration optimization models only consider economic and environmental benefits, with less attention to new energy consumption and system flexibility. Furthermore, there is a lack of a comprehensive value evaluation system for the multi-dimensional evaluation of the benefits brought by P2H equipment after the system is operational. Establishing an evaluation system that comprehensively considers these four dimensions as assessment criteria enables better utilization of renewable energy resources, enhances system flexibility and stability, and achieves comprehensive economic and environmental benefits. Meanwhile, in solving the P2H equipment capacity optimization configuration model, traditional mathematical solution methods, due to the higher-order nonlinearity of the objective function in integrated energy systems, have become less applicable. The use of metaheuristic algorithms can effectively address this issue. Furthermore, in multi-objective optimization

**Table 1. Review of recent relevant studies along with the present study.**

| Reference | Optimization Configuration Model | Objective Function | | | | | Solution Method |
|---|---|---|---|---|---|---|---|
| | | Renewable Energy Integration | Economic Efficiency | Flexibility | Environmental Benefits | Others | |
| [2] | Hydrogen Storage | | √ | | | | MILP |
| [3] | RSOC System | | √ | | | √ | PSO |
| [5] | PV-Hydrogen System | | √ | | | | Gurobi |
| [7] | PV/battery/ hydrogen system | | √ | | | | Gurobi |
| [8] | IES | | √ | | √ | | NSGA-II |
| [9] | IES | | √ | | √ | √ | NAGA-II |
| [10] | Energy Storage Planning | | √ | | | | Genetic quantum algorithm |
| [12] | IES | | √ | | √ | | MILP |
| [18] | Solar Power Generation | | | | | √ | BAPSO |
| [19] | Grid Hosting Capacity Model | | | | √ | √ | MPA-HBA |
| [22] | IES | √ | √ | | | | HMPSO |
| [23] | Diesel/PV/FC System | √ | √ | | | √ | MOCSA |
| This study | P2H | √ | √ | √ | √ | | IBBPSO |

problems, methods based on Pareto usually require trade-offs between objectives and generate non-dominated solution sets, which significantly increase computational complexity. However, by combining multiple objectives into a single objective using weight coefficients, the model can be effectively simplified, thus improving computational efficiency.Given these challenges, the main contributions of this paper are as follows:

(1) A four-dimensional optimization model is constructed considering new energy consumption, economic benefits, system flexibility, and environmental benefits. An improved backbone particle swarm optimization algorithm (IBBPSO) is used to solve the model. The performance of IBBPSO is compared with that of the traditional PSO algorithm and the backbone particle swarm optimization (BBPSO) algorithm to verify its superiority.

(2) A multi-dimensional value evaluation system based on the AHP -entropy weight method is constructed, including four first-level and eight second-level indicators, providing a relatively complete framework for the value evaluation of P2H equipment in IES. By comparing the value evaluation of P2H equipment in different capacity configurations, the appropriate P2H configuration is analyzed, and the evaluation system's systematicness, effectiveness, and practicability are proved.

The remainder of this article is organized as follows. Section 2 proposes the system structure and the value evaluation strategy structure diagram. Section 3 describes the mathematical model and the improved algorithm. Section 4 presents a multi-dimensional value evaluation system. Section 5 carries out the simulation and analysis. Finally, Section 6 summarizes this study.

## 2  Problem description

### 2.1  System structure

The structure of the IES with P2H is shown in Fig 1. The source-side equipment of the system, such as the wind turbine, cogeneration unit, and gas turbine unit, is responsible for providing

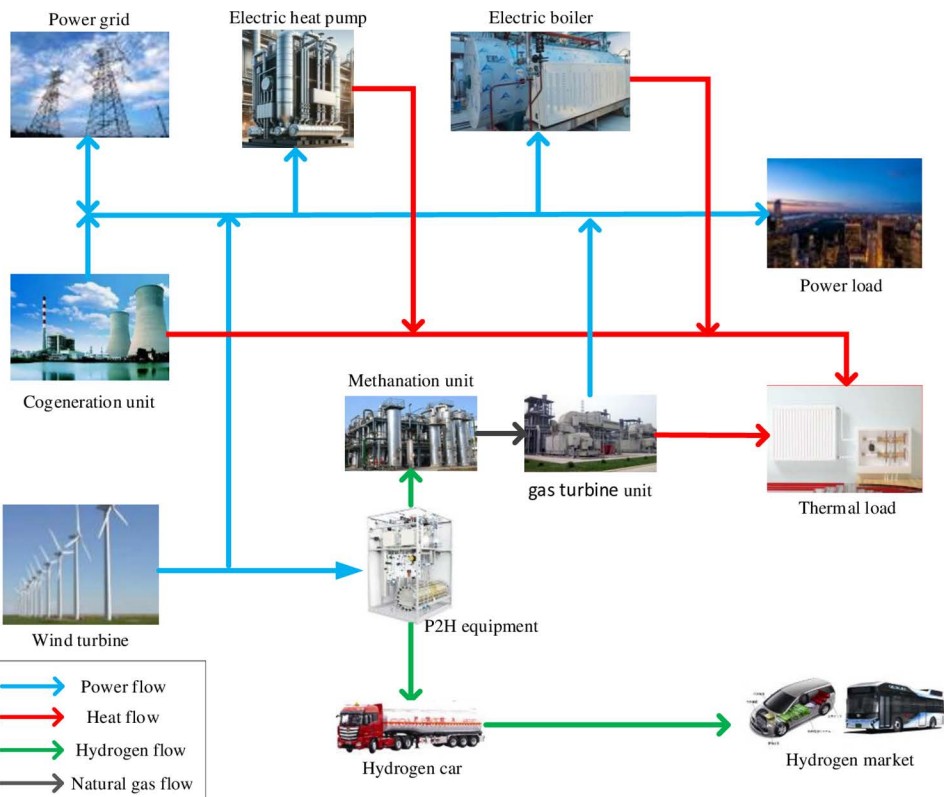

**Fig 1. Structure diagram of IES with P2H.**

the necessary energy for the entire system. The intermediate equipment includes P2H equipment, an electric boiler, and an electric heat pump, which promotes the effective absorption of abandoned wind and provides the necessary flexibility for the system. The load side comprises electric load, heat load, and hydrogen market, which meets the needs of residents 'lives and industrial production.

By using the technology of P2H, the IES can realize the efficient integration and utilization of various resources such as electricity, heat, and gas. The IES that employs P2H technology not only effectively lowers the wind curtailment rate but also achieves the secondary utilization of electric energy as hydrogen energy, thereby securing the system's economic benefits. Part of the hydrogen produced by the P2H equipment is methane to generate natural gas for the gas turbine unit used to generate electrical and thermal energy; the other part of hydrogen is transported to the hydrogen market through hydrogen tankers for industrial use of hydrogen or to meet other hydrogen energy needs.

## 2.2 Strategic structure

The strategic structure of the P2H equipment configuration is shown in Fig 2. To achieve energy conservation and emission reduction, maximize economic benefits, reduce wind curtailment rate and improve system flexibility of the IES, and determine the reasonable capacity configuration of P2H equipment in IES under these goals, a mathematical model considering new energy consumption, system flexibility, economy, and environmental benefits is constructed, and a comprehensive multi-dimensional value evaluation system is

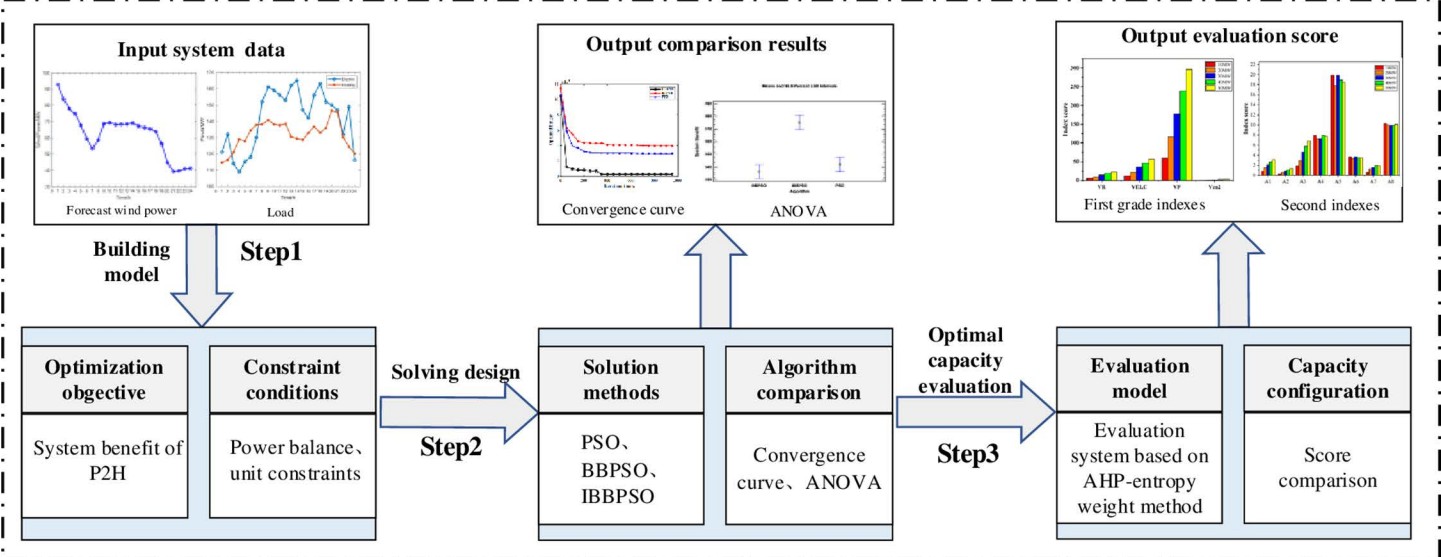

**Fig2. Value evaluation strategy structure of P2H equipment configuration in IES.**

proposed. After obtaining the optimal solution of the mathematical model, a comprehensive value evaluation of the P2H equipment with different capacity configurations in the IES is carried out. The specific steps are as follows:

**Step 1: Establish the mathematical model of the system.** The optimization model is established by considering the system power balance, generator set, and equipment output constraints. The objective function is to maximize the comprehensive benefits brought by the P2H equipment to the IES, including new energy consumption, flexibility, and economic and environmental benefits. The input data includes 24-hour electric load demand, heat load demand, and wind turbine forecast output on a typical day.

**Step 2: Design the solution algorithm.** IBBPSO is used to solve the mathematical model and compared with PSO and BBPSO. The effectiveness of the improved algorithm is verified by the output convergence curve and the results of ' multi-factor ANOVA '.

**Step 3: Build an evaluation system and value evaluation.** Based on the AHP-entropy weight method, a multi-dimensional value evaluation system is established. When the comprehensive benefits brought by the P2H equipment to the IES are optimal, the value of the P2H equipment with different capacity configurations is compared, and the scores of the first-level indicators and the second-level indicators are output to find the appropriate capacity configuration and verify the effectiveness of the value evaluation system.

## 3 Mathematical model

### 3.1 Objective functions

To maximize the comprehensive benefits brought by P2H equipment for IES, the new energy consumption benefits, hydrogen sales economic benefits, flexibility benefits, and environmental benefits brought by P2H equipment are comprehensively considered:

$$\max V = (\alpha_1 V_R + \alpha_2 V_{ELC} + \alpha_3 V_F + \alpha_4 V_{CO_2}), \tag{1}$$

where $V$ represents the comprehensive benefits provided by the P2H equipment; $V_R$ represents the economy brought by the P2H equipment to absorb the excess wind power of the wind farm, that is, the reduced cost of wind abandonment; $V_{ELC}$ represents the economic benefits of hydrogen sales; $V_F$ represents the flexibility benefit brought by the P2H equipment as a flexible resource; $V_{CO_2}$ represents the carbon treatment economy saved by reducing CO2 emissions from hydrogen-based synthetic natural gas. $a_1$, $a_2$, $a_3$, $a_4$ are the weight coefficients between different objectives, which are calculated by AHP(Section 4.2). $V_R$, $V_{ELC}$, $V_F$, $V_{CO2}$ satisfy the relationship:

$$\sum_{i=1}^{4} \alpha_i = 1,\ 0 \le \alpha_i \le 1, \tag{2}$$

$$\begin{cases} V_R = \lambda_{pel} \cdot P_{ELC} \\ V_{ELC} = \sum_{t=1}^{T} \lambda_{H_2} \dfrac{P_{elc,t} \cdot \Delta t}{\mu_h} \\ V_F = \sum_{t=1}^{T} [\sqrt{(P_{ELC,t}^U)^2 + (P_{ELC,t}^D)^2}] \Delta t \\ V_{CO_2} = \sum_{t=1}^{T} \lambda_{CO_2} V_{H_2} \rho_{CO_2} \end{cases} \tag{3}$$

where $P_{ELC}$ represents the total amount of excess wind power absorbed by the P2H equipment in the operation cycle of the wind farm; $P_{elc,t}$ represents the abandoned wind power consumed by the P2H equipment at time $t$; $P_{ELC,t}^U$ and $P_{ELC,t}^D$ respectively represent the upward flexibility and downward flexibility provided by the P2H equipment at time $t$; $V_{H_2}$ represents the total amount of hydrogen produced by the P2H equipment; $\lambda_{pel}$ represents the penalty coefficient of wind abandonment; $\lambda_{H_2}$ represents the selling price of hydrogen per unit volume; $\mu_h$ represents the abandoned wind power consumed by generating unit volume of hydrogen; $\lambda_{CO_2}$ represents the cost of treating a unit volume of CO2; $\rho_{CO_2}$ represents the coefficient of CO2 emission reduction per unit volume of hydrogen. Among them,

$$\begin{cases} P_{ELC,t}^D = \left| P_{elc,t} - P_{elc,\min} \right| \\ P_{ELC,t}^U = \left| P_{elc,\max} - P_{elc,t} \right|, \\ V_{H_2} = \sum_{t}^{T} \dfrac{P_{elc,t} \cdot \Delta t}{\mu_h} \end{cases} \tag{4}$$

where $P_{elc,\min}$ represents the minimum value of wind energy absorbed by P2H equipment; $P_{elc,\max}$ represents the maximum value of wind energy absorbed by P2H equipment.

**3.1.1 Decision variables.** The decision variables include the cogeneration unit output $P_{CHP}$ and $Q_{CHP}$, the electricity consumption power of the P2H equipment $P_{elc}$, the gas consumption of the gas turbine unit $P_{GT,g}$, the electrical energy consumption of the electric boiler $P_{EB}$ and the electric heat pump $P_{HP}$.

## 3.2 System constraints

(1) Electric power balance constraint:

$$P_{CHP,t} + P_{pw,t} + P_{GT,t} = P_{l,t} + P_{elc,t} + P_{EB,t} + P_{HP,t},\tag{5}$$

where $P_{CHP,t}$ represents the electric energy generated by the cogeneration unit at time $t$; $P_{pw,t}$ represents the electrical energy generated by the wind turbine at time $t$; $P_{GT,t}$ represents the electric energy generated by the gas turbine unit at time $t$; $P_{l,t}$ represents the power demand for active load at time $t$; $P_{EB,t}$ represents the electric energy consumed by the electric boiler at time $t$; $P_{HP,t}$ represents the electric energy consumed by the electric heat pump at time $t$.

(2) Thermal power balance constraint:

$$Q_{CHP,t} + Q_{EB,t} + Q_{HP,t} + Q_{GT,t} = Q_{l,t},\tag{6}$$

where $Q_{CHP,t}$ represents the heat energy generated by the cogeneration unit at time $t$; $Q_{EB,t}$ represents the heat energy generated by the electric boiler at time $t$; $Q_{HP,t}$ represents the heat energy generated by the electric heat pump at time $t$; $Q_{GT,t}$ represents the heat energy generated by the gas turbine unit at time $t$; $Q_{l,t}$ represents the power demand for the heat load at time $t$.

(3) Wind turbine constraint:

$$\underline{P_{pw}} \leq P_{pw,t} \leq \overline{P_{pw}},\tag{7}$$

where $\underline{P_{pw}}$ represents the wind turbine's minimum output; $\overline{P_{pw}}$ represents the wind turbine's maximum output.

(4) Power-to-hydrogen equipment constraint:

$$P_{elc,\min} \leq P_{elc,t} \leq P_{elc,\max},\tag{8}$$

(5) Cogeneration unit constraints:

$$P_{CHP,\min} \leq P_{CHP,t} \leq P_{CHP,\max},\tag{9}$$

$$Q_{CHP,\min} \leq Q_{CHP,t} \leq Q_{CHP,\max},\tag{10}$$

$$\max(C_v \cdot Q_{CHP,t} + P_{CHP,D}, C_m \cdot Q_{CHP,t} + P_{CHP,C}) \leq P_{CHP,t} \leq C_v \cdot Q_{CHP,t} + P_{CHP,A},\tag{11}$$

where $P_{CHP,\min}$ represents the minimum value of the electric output of the cogeneration unit; $P_{CHP,\max}$ represents the maximum power output of the cogeneration unit; $Q_{CHP,\min}$ represents the minimum value of the thermal output of the cogeneration unit; $Q_{CHP,\max}$ represents the maximum thermal output of the cogeneration unit. The electro-thermal coupling relationship

of the cogeneration unit is as follows Formula (11), where $C_V$, $P_{CHP,D}$, $C_m$, $P_{CHP,C}$ and $P_{CHP,A}$ are coupling parameters.

(6) Gas turbine unit output constraints:

$$P_{GT,t} = \eta_e P_{GT,g}, \tag{12}$$

$$Q_{GT,t} = \eta_h P_{GT,g}, \tag{13}$$

$$P_{GT,g}^{\min} \leq P_{GT,g} \leq P_{GT,g}^{\max}, \tag{14}$$

where $P_{GT,g}$ represents the gas consumption power of the gas turbine unit; $\eta_e$ represents the gas-to-electricity efficiency coefficient; $\eta_h$ represents the gas-to-heat efficiency coefficient; $P_{GT,g}^{\min}$ represents the minimum gas consumption power; $P_{GT,g}^{\max}$ represents the maximum gas consumption power.

(7) Electric heat pump constraints:

$$P_{HP,\min} \leq P_{HP,t} \leq P_{HP,\max}, \tag{15}$$

$$Q_{HP,t} = COP \cdot P_{HP,t}, \tag{16}$$

where $P_{HP,\max}$ and $P_{HP,\min}$ respectively represent the electric heat pump's maximum and minimum power consumption; $COP$ represents the power-to-heat coefficient.

(8) Electric boiler constraints:

$$Q_{EB,t} = \eta_{EB,e} \cdot P_{EB,t}, \tag{17}$$

$$P_{EB,\min} \leq P_{EB,t} \leq P_{EB,\max}, \tag{18}$$

where $\eta_{EB,e}$ represents the power-to-heat coefficient; $P_{EB,\max}$ and $P_{EB,\min}$ respectively represent the electric boiler's maximum and minimum power consumption.

## 3.3 Solution algorithm design

### 3.3.1 Presentation of PSO and BBPSO.
The traditional PSO topology is usually a global neighborhood topology. In this structure, each particle can communicate with all other particles in the swarm, meaning that all particles can directly obtain the position of the global optimum [33]. The advantage of this topology is that it can accelerate the search for the global optimum, but it may also lead to the algorithm easily getting trapped in local optima. The formula for particle position $x_i^{k+1}$ and velocity $v_i^{k+1}$ update is shown in Formula (19):

$$\begin{cases} v_i^{k+1} = \omega v_i^k + c_1 r_1 (P_i^k - v_i^k) + c_2 r_2 (P_g^k - v_i^k) \\ x_i^{k+1} = x_i^k + v_i^{k+1} \end{cases}, \tag{19}$$

where $k$ is the iteration number; $w$ is the inertia weight factor; $c_1$ and $c_2$ are learning factors; $P_i^k$ is the individual best position; $P_g^k$ is the global best position; $r_1, r_2 \sim U(0,1)$.

The BBPSO algorithm uses Gaussian distribution regarding the global guider and individual guider of particles to update the particle positions [34]. It does not require setting control parameters such as inertia weight and learning factors, making it a particle swarm optimization algorithm with fewer control parameters. The particle position update formula is shown as Formula (20):

$$\begin{cases} N(\dfrac{r_3 \cdot pbest(k) + (1-r_3) \cdot gbest(k)}{2}, |\, pbest(k) - gbest(k)\,|), \ if \ U(0,1) < 0.5 \\ gbest(k), \ else \end{cases} , \tag{20}$$

where $pbest(k)$ represents the individual guider, which is the best position each particle has found during the search process; $gbest(k)$ represents the global guider, which is the best position found by the entire particle swarm; $r_3 \sim U(0,1)$.

**3.3.2 Improved backbone particle swarm optimization algorithm.** In some cases, the BBPSO algorithm may still converge to the local optimal solution. To overcome this challenge and improve the global exploration ability of the algorithm in the early stage of search and the local refinement ability in the later stage of search, an improved BBPSO algorithm is introduced. The algorithm improves the particle position update formula in BBPSO to reduce the risk of premature convergence to the local optimal solution and ensure that a more comprehensive search space is explored [35]. The improved formula, as shown in Formula (21):

$$\begin{cases} x(k+1) = \begin{cases} N(\dfrac{w_1 \cdot pbest(k) + w_2 \cdot gbest(k)}{2}, w_* \cdot \sigma^2) + w_3, if \ U(0,1) < 0.5 \\ gbest(k), else \end{cases} \\ w_1 = 1 - 0.6\cos(\pi * k / T_{max}) \\ w_2 = 1 + 0.6\cos(\pi * k / T_{max}) \\ w_* = 0.6 + 0.4\cos(\pi * k / T_{max}) \\ w_3 = rand * (gbest(k) - pbest(k)) \\ \sigma^2 = |pbest(k) - gbest(k)| \end{cases} , \tag{21}$$

where $w_1$ represents the local search factor; $w_2$ represents the global search factor; $w_*$ represents the search range factor; $w_3$ represents the accelerating convergence factor.

In the early stage of the search, due to the small number of iterations, $w_1$ is smaller, while $w_2$ is larger, thus increasing the global search range; in the later stage of the search, $w_1$ is larger, while $w_2$ is smaller, the local search ability is enhanced, and the solution result is more accurate.

In the IBBPSO algorithm, the particle swarm is initialized first, and the fitness of each particle is evaluated to determine the individual and global optimal positions. Then, in each iteration, the particle position is updated by the above formula, and the boundary processing and fitness evaluation are performed to update the individual and global optimal positions until the maximum number of iterations or satisfactory fitness level is reached. The solving process of the IBBPSO algorithm is shown in Fig 3. The specific steps are as follows:

| |
|---|
| **Algorithm**: Improved backbone particle swarm optimization algorithm |
| **Input:** Particle swarm $x_i$ ($i$=1, 2, 3, …, $N$); Decision variable dimensions, $Dim$; Iteration times, $maxIters$; Particle swarm boundary, $up\_bound$ and $down\_bound$ |
| **Output:** Best solution found |

1 **For** $i$=1, 2, …, $N$ **do**

2 *Initialize $x_i$ and pbest*;

3 Fitness evaluation;

4 **End**

5 Select the best fitness *gbest*;

6 **While** *Iter* ≤ *maxIters* **do**

7 **For** i=1, 2, …, $N$ **do**

8 **If** rand(1) <0.5 **Then**

9 Update $x_i$ by (21);

10 **Else**

11 $x_i$=*gbest*;

12 **End**

13 **If** $x_i \geq up\_bound$ **Then**

14 $x_i = up\_bound$ ;

15 **End**

16 **If** $x_i \leq down\_bound$ **Then**

17 $x_i = down\_bound$ ;

18 **End**

19 Fitness evaluation and update *pbest*;

20 **End**

21 Select the best fitness *gbest*;

22 **End**

# 4 Multi-dimensional evaluation of capacity configuration of IES with P2H

## 4.1 Multi-dimensional evaluation system

The evaluation system of IES with P2H adopts a tree structure, aiming at comprehensively evaluating the P2H equipment and its application benefits in the IES. The evaluation indicators are divided into two main categories to ensure the comprehensiveness and depth of the evaluation:(1) First-level indicators: The main direction and scope of the evaluation are determined, which are used to summarize the main evaluation areas and key performance indicators. (2) Second-level indicators: Based on the framework of the first-level indicators, the second-level indicators are further refined and analyzed in depth, with precise calculation

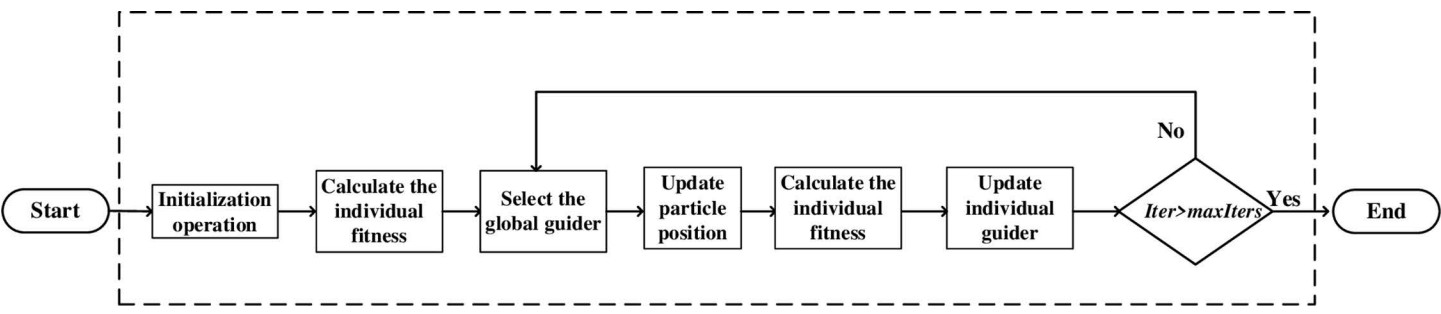

**Fig 3. Solving flow chart of the IBBPSO algorithm.**

methods and data sources to support the first-level indicators. The performance of P2H technology in IES is discussed and analyzed in depth by quantifying specific parameters.

**4.1.1 First-level indicators.** When constructing the value evaluation system, selecting the first-level indicators follows three principles: systematic, independent, and scientific, aiming to evaluate the benefits of P2H technology for IES from a broad perspective. The comprehensive value evaluation system has four first-level indicators: new energy consumption, economy, flexibility, and environmental benefits, as shown in Fig 4.

**4.1.2 Second-level indicators.** The second-level index is the basis of the whole evaluation system. It is the decomposition and quantification of the first-level index. It has four characteristics: measurability, clarity, consistency, and timeliness. It aims to evaluate the benefits of P2H technology for IES deeply. As shown in Fig 4, starting from the four first-level indicators, combined with the index selection characteristics, eight second-level indicators are selected, specifically:

(1)Flexibility

The flexibility margin $F_{f,s}$ is defined as the ratio of the flexibility provided by the P2H equipment in the system to the total flexibility during the operation cycle, as shown in Formula (22). The flexibility supply $F_{f,d}$ is defined as the ratio of the flexibility provided by the P2H equipment at time t to the total flexibility in the system at that time, as shown in Formula (23):

$$F_{f,s} = \sum_{t=1}^{T} \frac{\sum_{mh=1}^{Mh} \Delta F_{h,t}}{\Delta F_t} \times 100\%, \qquad (22)$$

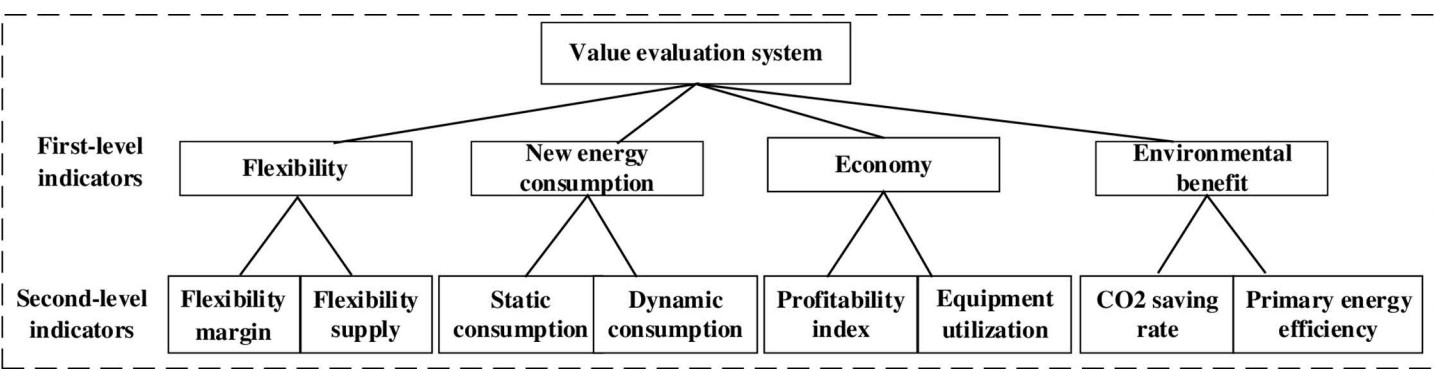

**Fig 4. Value evaluation system.**

$$F_{f,d} = \frac{\sum\limits_{mh=1}^{Mh} F_{h,t}}{\sum\limits_{k=1}^{M} F_{f,t}^{k}} \times 100\%, \tag{23}$$

where $\Delta F_{h,t}$ represents the flexibility provided by the P2H equipment per unit time; $\Delta F_t$ represents the flexibility provided by the system per unit time; $T$ represents the running cycle; $Mh$ represents the number of P2H equipment; $F_{h,t}$ represents the flexibility provided by the P2H equipment at time $t$; $F_f^k$ represents the flexibility provided by the kth flexibility resource in the system at time $t$; M represents the number of flexibility resources.

(2)New energy consumption

The static consumption rate $F_{w,s}$ is defined as the ratio of excess wind energy to original wind energy absorbed by the P2H equipment, as shown in Formula (24). The dynamic consumption rate $F_{w,d}$ is defined as the ratio of the square sum of the square of the wind energy change rate of the P2H equipment in the adjacent period to the maximum capacity of the P2H equipment, as shown in Formula (25):

$$F_{w,s} = \sum_{t=1}^{T} \frac{P_{elc,t}}{P_{res,t}^{cur} + P_{elc,t}} \times 100\%, \tag{24}$$

$$F_{w,d} = \frac{\sqrt{\dfrac{1}{T}\sum\limits_{t=1}^{T}[P_{elc,t} - P_{elc,t-1}]^2}}{P_{elc,\max}} \times 100\%, \tag{25}$$

where $P_{res,t}^{cur}$ represents the remaining wind power after the conversion of P2H at time $t$.

(3)Economy

The profitability index $I_{PI}$ is defined as the present value of the unit investment, that is, the ratio of the present value of the system value of all expected future hydrogen sales to the initial investment, as shown in Formula (26), and the profitability index should be greater than 1. The equipment utilization $D_{urd}$ is defined as the ratio of the total amount of wind power consumed by the P2H equipment during the operation cycle to the total power consumption of the P2H equipment at full load, as shown in Formula (27):

$$I_{PI} = \frac{1}{C_{AC}} \left[ \sum_{y=0}^{L_h} V_y (1+\rho)^{-y} \right], \tag{26}$$

$$D_{urd} = \frac{\sum\limits_{t=1}^{T} P_{elc,t}}{T \cdot P_{elc,\max}} \times 100\%, \tag{27}$$

where $V_y$ represents the hydrogen sales revenue on the yth day; $\rho$ represents the daily interest rate of the bank; $C_{AC}$ represents the initial investment, which is related to $P_{elc,\max}$ and the unit investment cost $\gamma_h$; $L_h$ represents the number of running days.

(4)Environmental benefit

The CO2 saving rate $F_{CO2}$ is defined as the ratio of the CO2 emissions saved by the system to the CO2 emissions when the thermal power unit is powered, as shown in Formula (28). The primary energy efficiency $F_\eta$ is defined as the ratio of the output of the cogeneration unit to the sum of the heat and power generation of the system during the operating cycle, as shown in Formula (29):

$$F_{CO2} = \frac{F_m^{elc}}{F_m + F_m^{elc}} \times 100\%, \tag{28}$$

$$F_\eta = \frac{W_{CHP}}{W_e + W_h} \times 100\%, \tag{29}$$

where $F_m^{elc}$ represents the amount of CO2 consumed by hydrogen methanation; $F_m$ represents the coal consumption of thermal power units under the same conditions; $W_{CHP}$ represents the total output of cogeneration units during the operation cycle; $W_e$ represents the total electricity production in the system during the operation cycle; $W_h$ represents the total heat production during the operation cycle.

## 4.2 Evaluation method

In the existing evaluation system, the distribution of weights usually adopts two methods: the subjective and the objective. The subjective weighting method adopts the AHP, relies on expert opinions, and is generally aligned with the system operation experience [36]. The objective weighting method uses the entropy weight method to evaluate the system's operating status based on the system's operating indicators, which is more aligned with the actual situation of the system [37]. The combination weighting strategy of AHP combined with the entropy weight method is adopted to integrate the advantages of subjective and objective evaluation methods and make up for their defects so that the weight distribution of the evaluation system is more in line with the needs of practical engineering applications.

The relative importance of subjective and objective weights is calculated, and the subjective and objective weight relationship coefficients $\varepsilon_i$ and $\delta_i$ of the final indicators are calculated as follows:

$$\begin{cases} \varepsilon_i = \dfrac{w_{si}}{w_{si} + w_{oi}}, \\ \delta_i = \dfrac{w_{oi}}{w_{si} + w_{oi}} \end{cases} \tag{30}$$

where $w_{si}$ represents the subjective weight of the second-level index of item $i$; and $w_{oi}$ represents the objective weight of the second-level index of item $i$. Then, combined with the obtained weight relationship coefficient, the final combined weight is calculated:

$$w_i = \frac{\varepsilon_i w_{si} + \delta_i w_{oi}}{\sum\limits_{i=1}^{m} (\varepsilon_i w_{si} + \delta_i w_{oi})}, \tag{31}$$

# 5 Simulation and analysis

## 5.1 Simulation setting

All experiments in this study were conducted using MatlabR2021a software in a 64-bit Windows 10 environment with a 1.80 GHz 8th Gen Intel(R) Core(TM) i7-8565U processor and 8 GB of RAM on a computer.

## 5.2 Original data

**5.2.1 Sample data.** The model is applied to an IES in Northeast China. The operation time of the IES with P2H is 24 h, and the operation interval is one h. Load power demand

and wind turbine output power are shown in Fig 5 and Fig 6. Experts are invited to score the relative importance of the secondary indicators. Based on these evaluations, a judgment matrix for the significance of the secondary indicators is derived to serve as the input data for the AHP, as shown in Table 2. Unit equipment parameters and other parameters are shown in Table 3 and Table 4.

In Table 1, A1, A2, A3, A4, A5, A6, A7, and A8 represent the flexibility supply, flexibility margin, static consumption rate, dynamic consumption rate, profitability index, equipment utilization rate, $CO_2$ emission reduction rate and primary energy utilization rate in turn.

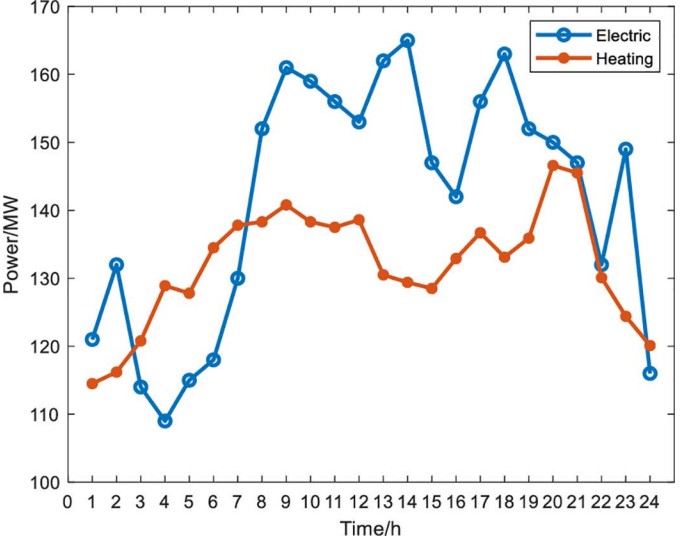

**Fig 5. System electric load and heat load curve.**

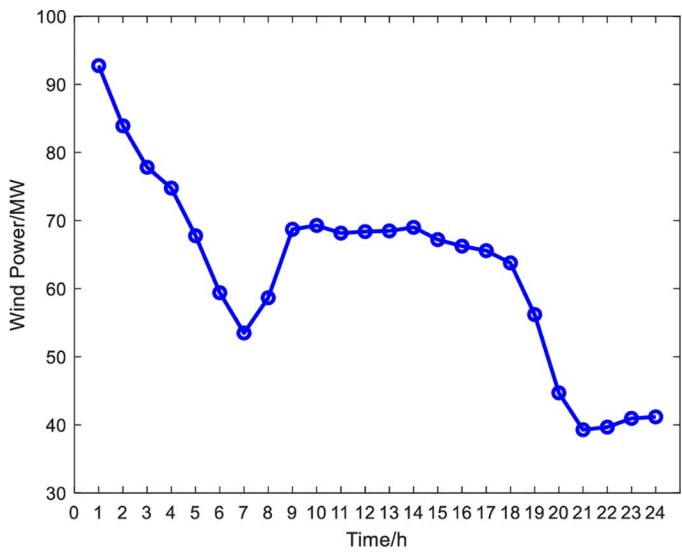

**Fig 6. Wind power forecast curve.**

**Table 2. Secondary index judgment matrix.**

| | $A_1$ | $A_2$ | $A_3$ | $A_4$ | $A_5$ | $A_6$ | $A_7$ | $A_8$ |
|---|---|---|---|---|---|---|---|---|
| $A_1$ | 1.0000 | 5.5724 | 0.9864 | 1.8517 | 2.5728 | 4.7134 | 5.8456 | 1.4543 |
| $A_2$ | 0.1795 | 1.0000 | 0.1770 | 0.3323 | 0.4617 | 0.8458 | 1.0490 | 0.2610 |
| $A_3$ | 1.0138 | 5.6495 | 1.000 | 1.8773 | 2.6084 | 4.7787 | 5.9265 | 1.4744 |
| $A_4$ | 0.5400 | 3.0093 | 0.5327 | 1.0000 | 1.3894 | 2.5455 | 3.1569 | 0.7854 |
| $A_5$ | 0.3887 | 2.1659 | 0.3834 | 0.7197 | 1.0000 | 1.8320 | 2.2721 | 0.5652 |
| $A_6$ | 0.2122 | 1.1822 | 0.2093 | 0.3929 | 0.5458 | 1.0000 | 1.2402 | 0.3085 |
| $A_7$ | 0.1711 | 0.9533 | 0.1687 | 0.3168 | 0.4401 | 0.8063 | 1.0000 | 0.2488 |
| $A_8$ | 0.6876 | 3.8318 | 0.6782 | 1.2733 | 1.7691 | 3.2411 | 4.0196 | 1.0000 |

**Table 3. Unit equipment parameters.**

| Unit | Parameter | Value | Unit | Parameter | Value |
|---|---|---|---|---|---|
| Cogeneration unit | $P_{CHP,e}^{min}$ /MW | 83.33 | Electric heat pump | $P_{HP}^{min}$ /MW | 0 |
| | $P_{CHP,e}^{max}$ /MW | 222 | | $P_{HP}^{max}$ /MW | 20 |
| | $P_{CHP,h}^{min}$ /MW | 50 | Electric boiler | $P_{EB}^{min}$ /MW | 0 |
| | $P_{CHP,h}^{max}$ /MW | 85 | | $P_{EB}^{max}$ /MW | 10 |
| Gas turbine unit | $P_{GT,g}^{min}$ /MW | 10 | P2H equipment | $P_{elc}^{min}$ /MW | 0 |
| | $P_{GT,g}^{max}$ /MW | 50 | | $P_{elc}^{max}$ /MW | 10/20/30/40/50 |

**Table 4. System-related parameters.**

| Parameter | Value | Parameter | Value | Parameter | Value | Parameter | Value |
|---|---|---|---|---|---|---|---|
| $C_V$ | 0.15 | $\eta_e$ | 0.375 | $COP$ | 3.2 | $\lambda_{pel}$ | 0.125 |
| $P_{m_{CHP},D}$ | 100 | $\eta_h$ | 0.625 | $\mu_h$ | 4.5 | $\lambda_{CO2}$ | 0.312 |
| $C_m$ | 0.75 | $\eta_{EB,e}$ | 0.9 | $\rho_{CO2}$ | 0.25 | $\rho$ | 0.0275 |
| $P_{m_{CHP},C}$ | 0 | $\gamma_h$ | 3.5 | $P_{m_{CHP},A}$ | 200 | $\lambda_{H_2}$ | 2.7 |

### 5.2.2 Parameter settings. (1) Algorithm parameters

The key algorithm parameters that need to be set for the IBBPSO algorithm include population size (Popsize), number of iterations (Iteration), and penalty factor (r). The Taguchi method is used to examine the impact of different levels of algorithm parameters on the solution. Each parameter is set to three levels, and an $L_{16}(3^3)$ -orthogonal experimental table is established, as shown in Table 5. Each experimental group runs 10 times, and the overall evaluation normalized value method is used to calculate the total evaluation normalized value (GM) to assess the parameters. The calculation formula is as follows:

$$\begin{cases} d_i = \dfrac{f_i - f_i^{max}}{f_i^{min} - f_i^{max}} \\ GM = \sqrt[k]{(d_1 d_2 ... d_k)} \end{cases} \tag{32}$$

**Table 5. $L_{16}(3^3)$ -orthogonal experiment table and results.**

| No. | Parameters | | | GM |
|---|---|---|---|---|
| | Popsize | Iteration | r | |
| 1 | 50 (Level 1) | 500 (Level 1) | 100 (Level 1) | 0.4522 |
| 2 | 50 | 1000 (Level 2) | 200 (Level 2) | 0.4604 |
| 3 | 50 | 1500 (Level 3) | 300 (Level 3) | 0.4541 |
| 4 | 100 (Level 2) | 500 | 200 | 0.4244 |
| 5 | 100 | 1000 | 300 | 0.4425 |
| 6 | 100 | 1500 | 100 | 0.4482 |
| 7 | 150 (Level 3) | 500 | 300 | 0.4023 |
| 8 | 150 | 1000 | 100 | 0.4331 |
| 9 | 150 | 1500 | 200 | 0.431 |

The trend of parameter factor level changes is shown in Fig 7. From the figure, it can be seen that the algorithm performs best when Popsize = 100, Iteration = 1000, and r = 100.

(2) Weight coefficients between different objectives

The determination of the weight coefficients $a_1, a_2, a_3, a_4$ in Formula (1) is based on the importance judgment matrix in Table 2, calculated using the AHP method, and then derived through sensitivity analysis. As shown in Fig 8, the analysis results indicate that the system achieves optimal performance $a_1 = 0.3706, a_2 = 0.1433, a_3 = 0.2813, a_4 = 0.2048$.

## 5.3 Capacity optimization configuration of IES with P2H

**5.3.1 Model solving.** According to the load demand curve, wind power forecast curve, the characteristics of the unit, and the capacity of the P2H equipment, to analyze the reasonable capacity configuration in detail, the capacity of the P2H equipment is set to 5 different levels: 10MW, 20MW, 30MW, 40MW, and 50MW. To maximize the comprehensive benefits provided by the P2H equipment for the IES, the IBBPSO algorithm is used to solve the model. To avoid

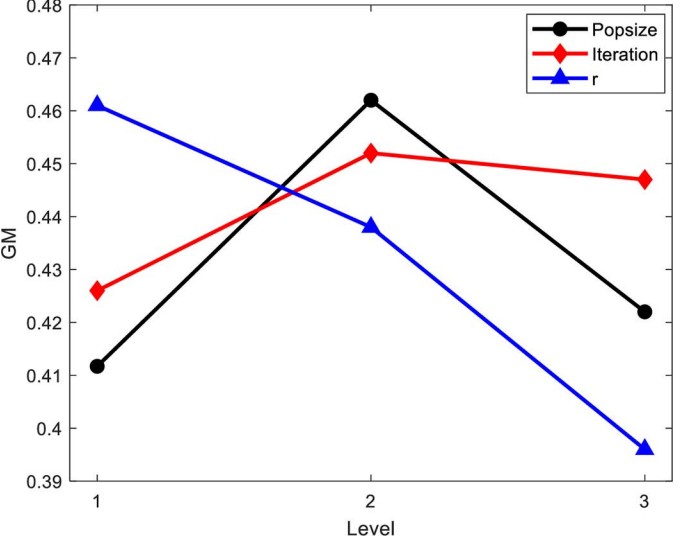

**Fig 7. Algorithm factor level change curve.**

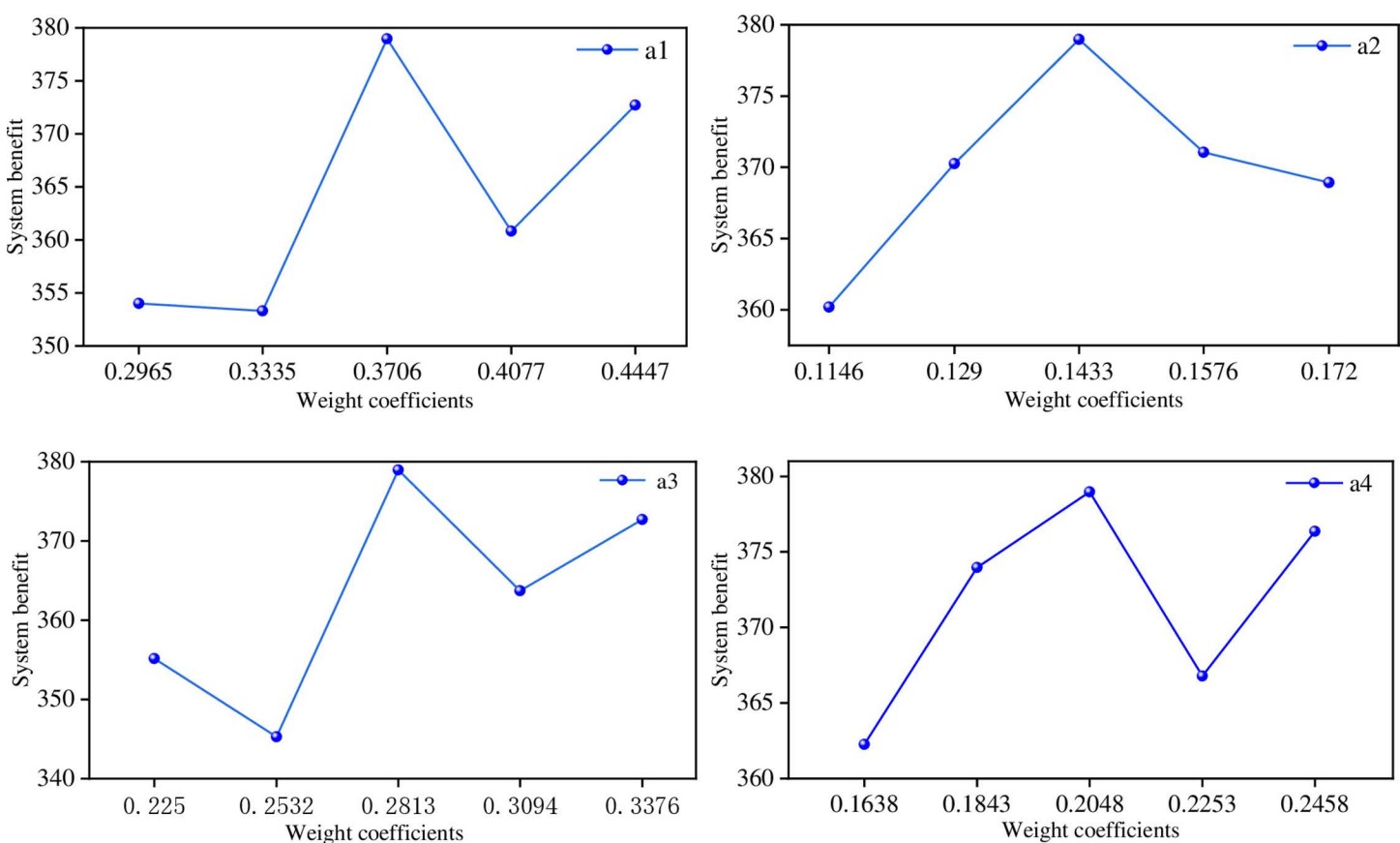

**Fig 8. Weight coefficients change curve.** (a) Weight coefficient of a1. (b) Weight coefficient of a2. (c) Weight coefficient of a3. (d) Weight coefficient of a4.

the influence of random errors, the comparison results of comprehensive benefits (V) under different capacity configurations are obtained by simulating ten times and taking the mean value. These benefits include four key first-level indicators: new energy consumption benefit ($V_R$), flexibility benefit ($V_F$), hydrogen sales economic benefit ($V_{ELC}$), and environmental benefit ($V_{CO2}$). The score comparison results of the first-level indicators are detailed in Table 6.

According to the results shown in Table 6, with the increase of the capacity of the P2H equipment, the comprehensive benefits brought by the P2H equipment to the integrated energy system and the benefits of each level of indicators are gradually increasing.

To verify the superiority of the IBBPSO algorithm in solving the comprehensive benefits brought by the P2H equipment to the system, it is compared with the traditional PSO algorithm, BBPSO algorithm, Grey Wolf Optimizer (GWO) and Whale Optimization Algorithm

**Table 6. First-level index results of different capacity configurations of P2H equipment.**

|  | 10MW | 20MW | 30MW | 40MW | 50MW |
|---|---|---|---|---|---|
| $V_R$ | 4.699 | 8.483 | 14.095 | 17.980 | 21.946 |
| $V_F$ | 59.829 | 116.301 | 177.399 | 238.184 | 296.230 |
| $V_{ELC}$ | 11.987 | 21.640 | 35.958 | 45.868 | 55.987 |
| $V_{CO2}$ | 0.489 | 0.883 | 1.467 | 1.872 | 2.285 |
| V | 77.004 | 147.306 | 228.920 | 303.904 | 376.447 |

(WOA). To specify the application scenario, the IES with a P2H capacity of 50 MW is selected as an example. To simplify the problem and facilitate the solution, the penalty function method deals with the equality constraints in the model. The comparison results are shown in Table 7.

The numerical results in Table 7 highlight the superior performance of the proposed IBB-PSO algorithm compared to other algorithms. Specifically, the IBBPSO algorithm achieves the highest comprehensive benefits V of 376.447, which is 11.09% and 9.8% higher than the BBPSO and PSO algorithms, respectively. Compared to the GWO and WOA algorithms, the comprehensive benefits V of the IBBPSO algorithm are 33.57% and 17.7% higher, respectively, demonstrating its excellent optimization capability across multiple objectives. In terms of flexibility benefits $V_F$, the IBBPSO algorithm achieves a value of 296.229, which is 14.79% and 9.21% higher than BBPSO and PSO, respectively, and 41.87% and 54.74% higher than GWO and WOA, respectively. Furthermore, the IBBPSO algorithm demonstrates significant advantages in renewable energy consumption benefits $V_R$ and environmental benefits $V_{CO2}$, highlighting its effectiveness in balancing economic, environmental, and flexibility objectives.

**5.3.2 Algorithm Comparison Analysis.** Fig 9 shows the convergence curve comparison diagram of the algorithm—the significant advantages of the IBBPSO algorithm in convergence performance. Compared with the PSO, BBPSO, GWO and WOA, the convergence speed is faster, and the convergence result is better. The ' multi-factor ANOVA ' is used to analyze the comprehensive benefit results with a 95% confidence interval to compare the influence of different algorithms on solving the optimization model. The results are shown in Fig 10. The

**Table 7. Numerical results under different algorithms.**

| Algorithms | V | $V_R$ | $V_F$ | $V_{ELC}$ | $V_{CO2}$ | runtime |
|---|---|---|---|---|---|---|
| PSO | 342.856 | 19.579 | 271.290 | 49.948 | 2.038 | 3.831s |
| BBPSO | 338.857 | 22.139 | 257.936 | 56.478 | 2.305 | 2.996s |
| GWO | 281.971 | 12.438 | 208.681 | 59.7 | 1.150 | 9.301s |
| WOA | 319.798 | 21.78 | 191.461 | 104.544 | 2.0134 | 3.641 |
| IBBPSO | 376.447 | 21.946 | 296.229 | 55.987 | 2.285 | 4.625s |

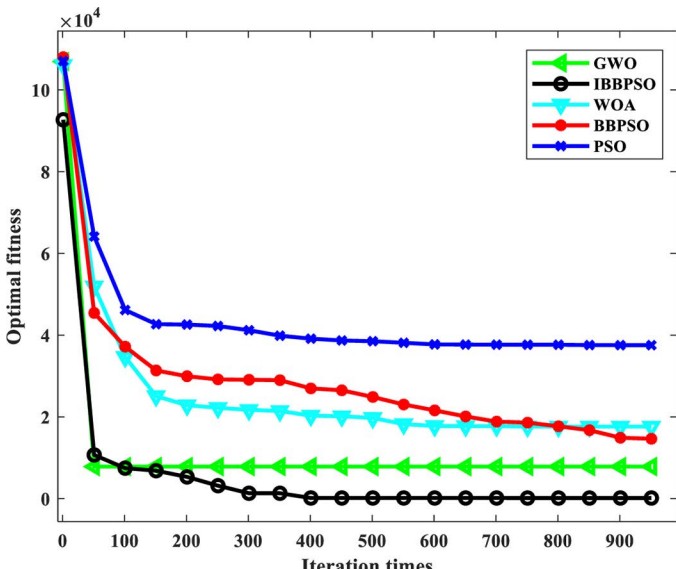

**Fig 9. Algorithm convergence curve.**

comprehensive benefit obtained by the IBBPSO algorithm is significantly higher than that of the other two algorithms, which proves its superior optimization ability.

To compare the result errors of different algorithms in solving this model, box plots were used to statistically analyze the results of 10 runs for each algorithm, as shown in Fig 11. The data distribution indicates that the median of the IBBPSO algorithm is closest to the mean and significantly higher than those of other algorithms, suggesting that this algorithm exhibits greater stability and minimal error.

Table 8 is for the time complexity analysis of the IBBPSO algorithm and PSO algorithm. The runtime results show that the average runtime of the IBBPSO algorithm is 4.332s, slightly higher than the 3.658s of the PSO algorithm, representing an 18.4% increase. This increase is attributed to the more refined particle position adjustment strategy and enhanced global-local search mechanism in the IBBPSO algorithm, which increase the computational complexity of each iteration.

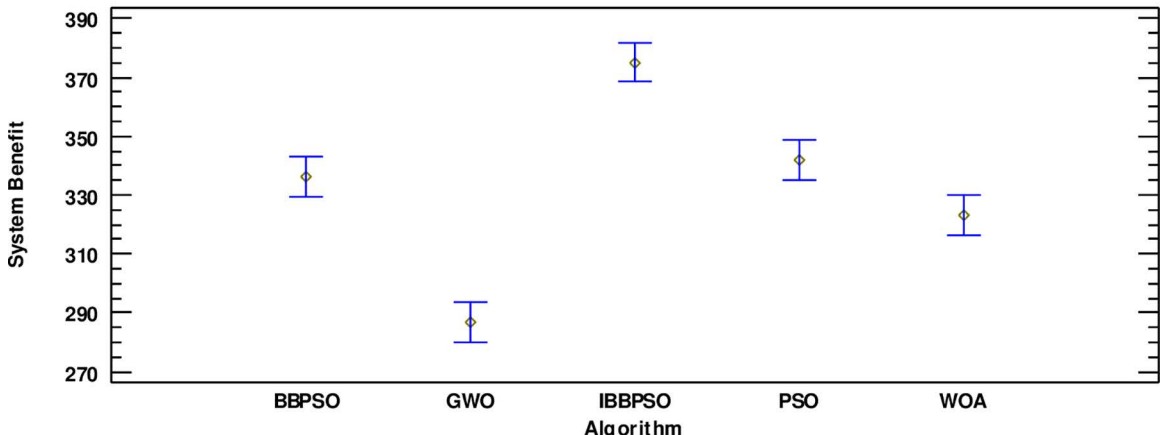

**Fig 10. Comparison of system benefits of different algorithms.**

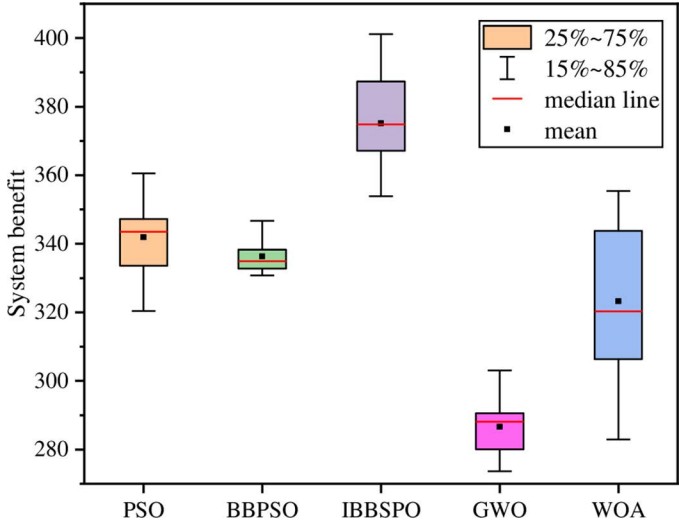

**Fig 11. Box plots of different algorithms.**

**Table 8. Comparison of runtime.**

| Times | Algorithm | 1 | 2 | 3 | 4 | 5 | 6 | 7 | 8 | 9 | 10 | Mean |
|---|---|---|---|---|---|---|---|---|---|---|---|---|
| Run time/s | IBBPSO | 4.392 | 4.313 | 4.273 | 4.328 | 4.217 | 4.462 | 4.366 | 4.298 | 4.246 | 4.426 | 4.332 |
| | PSO | 3.561 | 3.624 | 3.592 | 3.646 | 3.667 | 3.540 | 3.852 | 3.584 | 3.913 | 3.600 | 3.658 |

However, the improvements in solution quality and faster convergence speed demonstrate that this trade-off is reasonable, highlighting the superiority and stability of the IBBPSO algorithm.

**5.3.3 Qualitative analysis of different algorithms.** To evaluate the stability of the IBBPSO algorithm, benchmark comparisons are conducted using the Hartmann 4-D, Schaffer F7, and Kowalik and Osborne test functions. The expressions of the benchmark functions and their associated parameters are shown in the Table 9, with a population size of Popsize = 50 and a maximum number of iterations Iteration = 500. The corresponding function plots and simulation results are presented in Figs 12–14. The convergence curves of different algorithms

**Table 9. Benchmark function.**

| Test function | Formula | lb | ub | dim |
|---|---|---|---|---|
| Hartmann 4-D | $f(x) = -\sum_{i=1}^{4} c_i \exp\left(-\sum_{j=1}^{4} a_{ij}(x_j - p_{ij})^2\right)$ | 0 | 1 | 6 |
| Schaffer F7 | $f(x) = \sum_{i=1}^{5}\left((x_i - a_{SH}(i,:))\cdot(x_i - a_{SH}(i,:))' + c_{SH}(i)^{-1}\right)$ | 0 | 10 | 4 |
| Kowalik and Osborne | $f(x) = \sum_{i=1}^{11}\left(a_i - \frac{x_1(b_i^2 + x_2 b_i)}{b_i^2 + x_3 b_i + x_4}\right)^2$ | -5 | 5 | 4 |

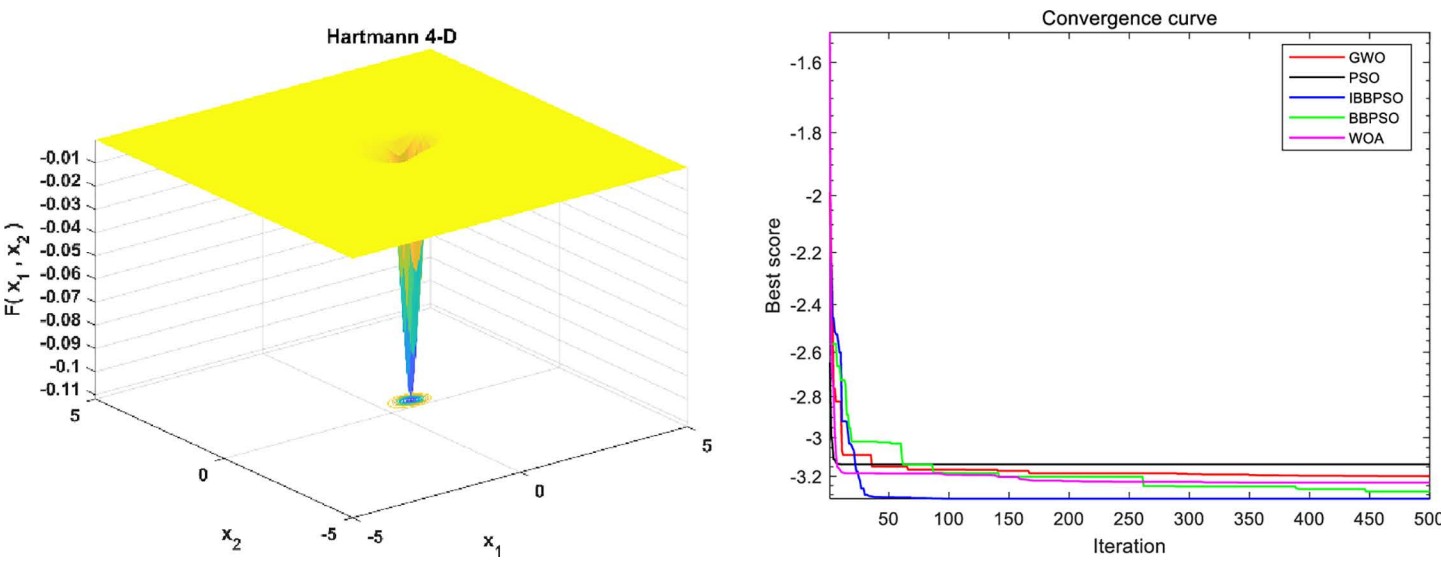

**Fig 12. Image and optimization comparison diagram for the Hartmann 4-D function.** (a) Hartmann 4-D three-dimensional function image. (b) Algorithm convergence curve comparison.

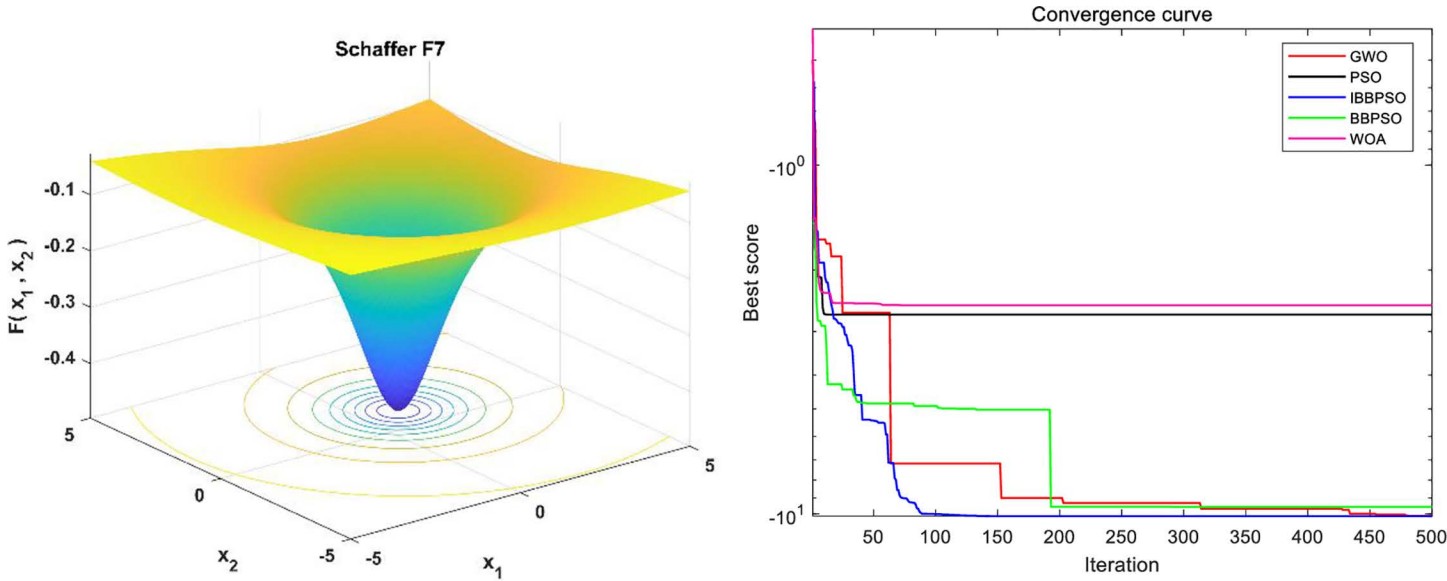

**Fig 13. Image and optimization comparison diagram for the Schaffer F7 function.** (a) Schaffer F7 three-dimensional function image. (b) Algorithm convergence curve comparison.

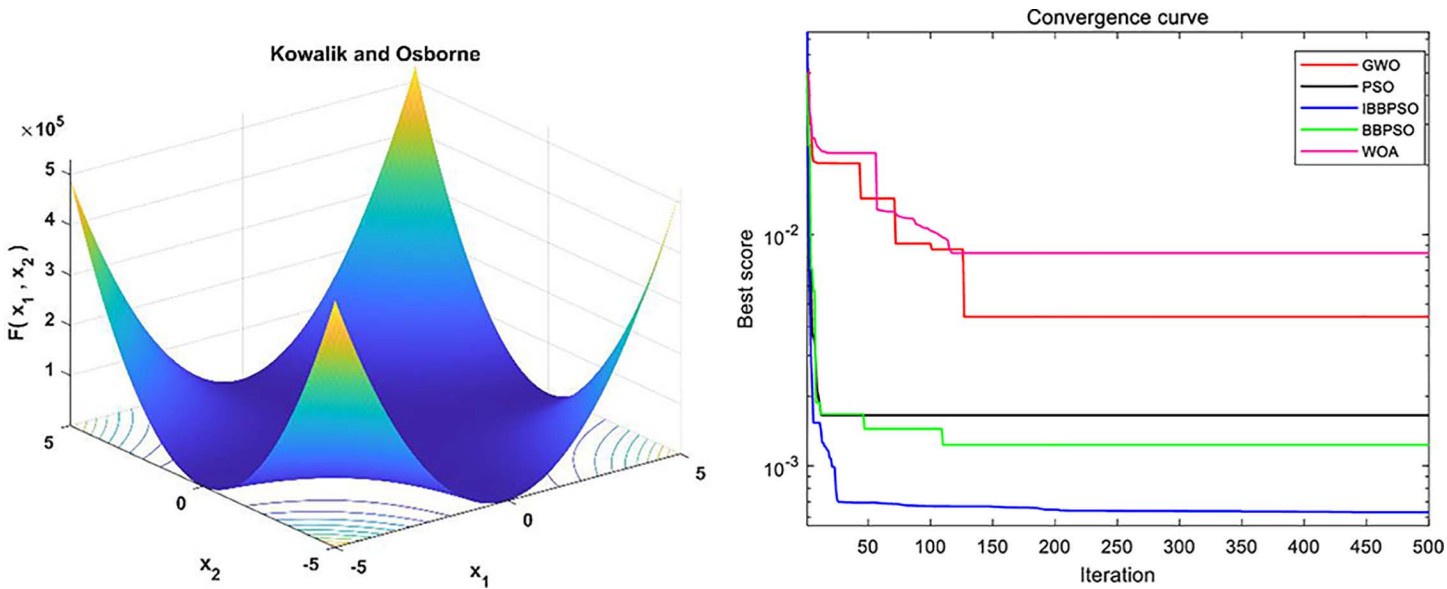

**Fig 14. Image and optimization comparison diagram for the Kowalik and Osborne function.** (a) Kowalik and Osborne three-dimensional function image. (b) Algorithm convergence curve comparison.

demonstrate that the IBBPSO algorithm exhibits significantly superior convergence speed and final results compared to the other algorithms.

### 5.4 Multi-dimensional value evaluation of P2H equipment configuration

To accurately evaluate the rational allocation of P2H equipment in the IES, achieve the goals of energy saving and emission reduction, improving system flexibility and economy, and

reducing the rate of abandoned wind, the multi-dimensional value evaluation of the role of P2H equipment in the IES is made when the optimization model is optimal. The data obtained when solving the optimization model in the previous section are used as the initial data of this section. And the secondary indexes of the IES with P2H under different capacity configurations are calculated according to the secondary index calculation formula (22-29), as shown in Table 10, and the first-level indicator scores are shown in Fig 15.

From the first-level indicators in Fig 15 the P2H equipment with a 50MW capacity configuration has a better value. According to the data in Table 10, with the increase of the capacity of the P2H equipment, the $CO_2$ emission reduction capacity in the integrated energy system is significantly improved. At the same time, the flexibility index has improved considerably. The larger the capacity configuration, the higher the adjustment capacity of the P2H equipment can provide for the IES, thereby enhancing the system's adaptability to the fluctuation of renewable energy.

To improve the accuracy and adaptability of the evaluation values, the subjective, objective, and final combination weights of each second-level indicator are calculated by combining the judgment matrix in Table 2 and the calculation results of each indicator in Table 10 according

**Table 10. Second-level evaluation indicators under different capacity configurations.**

|  | 10M | 20MW | 30MW | 40MW | 50MW |
|---|---|---|---|---|---|
| $A_1$ | 4.188 | 8.141 | 11.378 | 14.395 | 16.799 |
| $A_2$ | 3.393 | 6.603 | 9.266 | 12.298 | 15.214 |
| $A_3$ | 10.114 | 15.979 | 24.841 | 30.806 | 36.697 |
| $A_4$ | 59.533 | 54.712 | 55.038 | 59.415 | 58.017 |
| $A_5$ | 2.074 | 1.872 | 2.074 | 1.984 | 1.937 |
| $A_6$ | 49.053 | 44.277 | 49.049 | 46.925 | 46.822 |
| $A_7$ | 7.685 | 17.346 | 21.427 | 27.446 | 27.061 |
| $A_8$ | 60.966 | 59.678 | 59.327 | 59.232 | 60.056 |

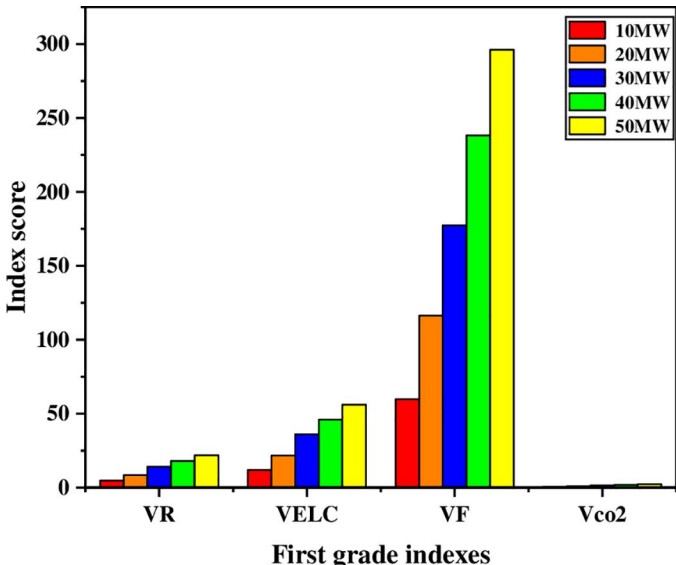

**Fig 15. First-level evaluation indicators under different capacity configurations.**

to the AHP-entropy weight method and Formula (30-31). The results are summarized in Table 11.

To ensure the consistency of the evaluation system and the convenience of comparison, all indicators are expressed in percentage form. Specific to the profitability index, it is standardized by multiplying it by 100%. The comprehensive evaluation scores of the IES with P2H under different capacity configurations are calculated by comprehensively using the index data provided in Table 10 and the combined weights in Table 11. The detailed results are shown in Table 12.

According to the data in Table 12, the comprehensive score is the highest when the P2H equipment in the system is configured at 50 MW, followed by the comprehensive score at 40 MW, and the comprehensive score is the worst at 10 MW. The index score histogram is drawn, as shown in Fig 16. Regarding flexibility and new energy consumption capacity, the P2H equipment in the system is significantly better than other capacity configurations when configured with a capacity of 50 MW. In comparison, 10 MW and 30 MW configurations are better than the profitability index. The CO2 emission reduction rate is the best when the capacity configuration is 40 MW, and the utilization rate of equipment and the primary energy utilization rate are less different in different capacity configurations. Therefore, the IES with solid flexibility and absorptive capacity is equipped with 30 MW P2H equipment to improve the profitability of the system; the IES with poor environmental benefits is equipped with 40 MW P2H equipment to enhance environmental benefits; the balanced system selects the capacity configuration with the best comprehensive score to improve the comprehensive performance.

Compared with the single index evaluation, the multi-dimensional value evaluation system can more comprehensively reflect the multi-faceted impact of P2H equipment on the IES,

**Table 11. Subjective and objective weights and combined weights of indicators.**

| First level | Second level | Subjective weight | Objective weight | Combined weight |
|---|---|---|---|---|
| $V_F$ | $A_1$ | 0.2385 | 0.1082 | 0.1828 |
| | $A_2$ | 0.0428 | 0.1155 | 0.0885 |
| $V_R$ | $A_3$ | 0.2418 | 0.1210 | 0.1862 |
| | $A_4$ | 0.1288 | 0.1567 | 0.1331 |
| $V_{ELC}$ | $A_5$ | 0.0927 | 0.1121 | 0.0954 |
| | $A_6$ | 0.0506 | 0.0961 | 0.0743 |
| $V_{CO2}$ | $A_7$ | 0.0408 | 0.0937 | 0.0717 |
| | $A_8$ | 0.1640 | 0.1967 | 0.1680 |

**Table 12. Comprehensive evaluation scores under different capacity configurations.**

| | 10MW | 20MW | 30MW | 40MW | 50MW |
|---|---|---|---|---|---|
| $A_1$ | 0.766 | 1.488 | 2.080 | 2.631 | 3.071 |
| $A_2$ | 0.300 | 0.584 | 0.820 | 1.088 | 1.346 |
| $A_3$ | 1.883 | 2.975 | 4.625 | 5.736 | 6.833 |
| $A_4$ | 7.924 | 7.282 | 7.326 | 7.908 | 7.722 |
| $A_5$ | 19.783 | 17.857 | 19.781 | 18.924 | 18.480 |
| $A_6$ | 3.645 | 3.290 | 3.644 | 3.487 | 3.479 |
| $A_7$ | 0.551 | 1.244 | 1.536 | 1.968 | 1.940 |
| $A_8$ | 10.246 | 10.026 | 9.967 | 9.951 | 10.089 |
| synthesis score | 25.512 | 27.068 | 30.197 | 32.959 | 34.666 |

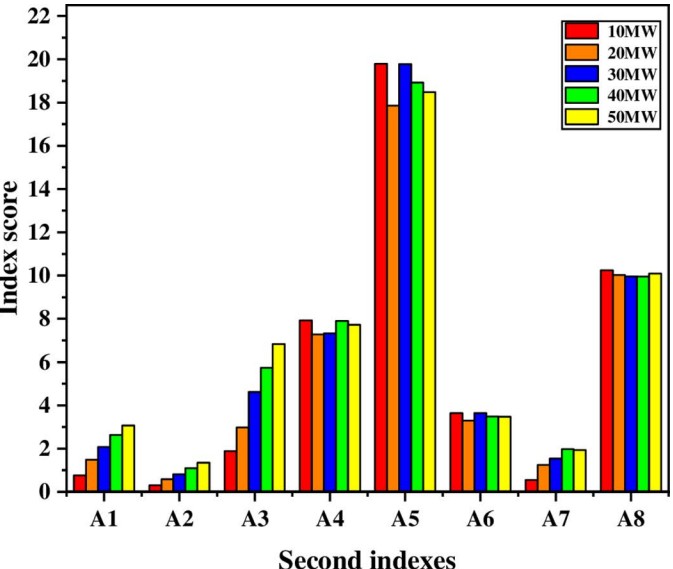

**Fig 16. Scores of each index under different capacity configurations.**

which is helpful to guide the reasonable capacity allocation of P2H equipment in the IES and verify the rationality, effectiveness, and practicability of the comprehensive evaluation system.

## 6 Conclusion

A mathematical model is proposed that considers new energy consumption, system flexibility, and economic and environmental benefits. The IBBPSO algorithm is used to solve the model. A multi-dimensional evaluation system is established, and the configuration of P2H equipment in the IES is evaluated and compared when the optimization model is optimal. Through in-depth analysis, the following main conclusions are drawn:

(1) The IBBPSO algorithm introduced global and local search factors to expand the search range, prevent the search results from falling into local optimum, and significantly enhance the algorithm's optimization ability. When solving the optimization configuration model, the IBBPSO algorithm yields a comprehensive benefit of 376.447. Compared to PSO, IBBPSO, GWO, and WOA algorithms, it improves by 9.8%, 11.09%, 33.57%, and 17.7%, respectively.

(2) The constructed multi-dimensional value evaluation system uses the combined weight of the AHP-entropy weight method to avoid the mutual interference between evaluation indicators and successfully identifies the reasonable configuration of P2H equipment in the IES when the comprehensive benefits brought by the P2H equipment to the system are the largest. The experiment shows that the highest comprehensive value is achieved when the capacity of the P2H equipment is 50 MW.

The limitations of this study primarily lie in the insufficient representativeness of the data, simplified model assumptions, and the computational efficiency of the algorithm, which may affect its applicability in different regions and larger-scale systems. Future research could enhance flexibility evaluation by expanding data sources, integrating various types of information, implementing dynamic assessment and real-time optimization, and introducing multi-objective optimization and intelligent scheduling methods such as reinforcement

learning. These advancements would improve the accuracy and timeliness of flexibility assessments, better address the complex and changing demands of power systems, and promote efficient and sustainable operation of power grids.

## Author contributions

**Conceptualization:** Xiaoyi Qian.

**Data curation:** Tianhe Sun, Xiaoyi Qian.

**Formal analysis:** Kuiyuan Pan.

**Funding acquisition:** Tianhe Sun.

**Methodology:** Kuiyuan Pan, Xinfu Pang.

**Project administration:** Xiaoyi Qian.

**Software:** Kuiyuan Pan, Tianhe Sun.

**Supervision:** Xinfu Pang.

**Validation:** Kuiyuan Pan.

**Visualization:** Xinfu Pang.

**Writing – original draft:** Kuiyuan Pan.

**Writing – review & editing:** Xinfu Pang.

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
