## [Decision Letter · Decision Letter 0]

26 Dec 2024

PONE-D-24-55307Capacity optimization configuration and multi-dimensional value evaluation of integrated energy system with power-to-hydrogenPLOS ONE

Dear Dr. Sun,

Thank you for submitting your manuscript to PLOS ONE. After careful consideration, we feel that it has merit but does not fully meet PLOS ONE’s publication criteria as it currently stands. Therefore, we invite you to submit a revised version of the manuscript that addresses the points raised during the review process.

We look forward to receiving your revised manuscript.

Kind regards,

Joy Nondy, Ph. D.

Academic Editor

PLOS ONE

Journal Requirements:

Reviewers' comments:

Reviewer's Responses to Questions

**Comments to the Author**

1. Is the manuscript technically sound, and do the data support the conclusions?

Reviewer #1: Yes

Reviewer #2: Yes

2. Has the statistical analysis been performed appropriately and rigorously? 

Reviewer #1: Yes

Reviewer #2: Yes

3. Have the authors made all data underlying the findings in their manuscript fully available?

Reviewer #1: Yes

Reviewer #2: Yes

4. Is the manuscript presented in an intelligible fashion and written in standard English?

Reviewer #1: Yes

Reviewer #2: Yes

5. Review Comments to the Author

Reviewer #1: The study proposes a multi-dimensional value evaluation system and optimization model, utilizing the IBBPSO algorithm and AHP-entropy weight method, to comprehensively assess and optimize power-to-hydrogen (P2H) equipment configurations in integrated energy systems, addressing flexibility, energy consumption, economic, and environmental benefits. The topic can be interesting. The comments are listed as follows to improve the quality of the paper.

1) The paper uses the Analytic Hierarchy Process (AHP) to determine the weight coefficients among different objectives. How can the issue of excessive conservatism be mitigated? Please provide additional explanations.

2) As stated in the paper, nonlinear problems are typically transformed into convex problems to achieve accurate global optimal solutions. In this context, the work “enabling high-efficiency economic dispatch of hybrid ac/dc networked microgrids: steady-state convex bi-directional converter models” should be discussed and cited in the literature review to enhance the quality of the paper.

3) Why not use more advanced Pareto-based methods to handle the multi-objective optimization problem? Please provide further explanations.

4) Integrated energy systems usually involve a high proportion of renewable energy, and renewable energy generation is inherently uncertain. A discussion on this aspect should be added to the paper, particularly regarding robust optimization methods, such as “an iteration-free hierarchical method for the energy management of multiple-microgrid systems with renewable energy sources and electric vehicles.” It is recommended that the authors elaborate on this and include the relevant references.

5) What is the motivation for using a meta-heuristic algorithm to solve the model rather than established solvers like CPLEX or Gurobi? Please clarify this choice.

6) Regarding Table 5, how can the results obtained by the proposed algorithm be demonstrated to outperform those of other algorithms? Please provide additional explanations.

Reviewer #2: This paper proposes a multi-dimensional value evaluation system to thoroughly assess P2H equipment with different capacity configurations in the integrated energy system. Additionally, an improved backbone particle swarm optimization algorithm is introduced to solve the proposed model. While the topic is interesting and timely, several points require clarification and improvement to enhance the quality and impact of the paper:

1. Please include the main numerical results in both the Abstract and Conclusion sections to summarize key findings clearly.

2. Compare the work presented in this paper with other studies in the literature, preferably in a tabular format, to highlight differences and improvements.

3. The introduction needs to be enhanced by discussing more recent meta-heuristic algorithms. Consider referencing studies such as: https://doi.org/10.1016/j.asej.2022.102092;
https://doi.org/10.1016/j.egyr.2022.04.066;
https://doi.org/10.1038/s41598-024-76410-0

4. Provide a detailed explanation of all algorithm parameters. Clarify why and how specific values were chosen and discuss how varying these values impacts the algorithm's overall performance.

5. Computational complexity is an important factor in any optimization algorithm and it is suggested that the authors add the computational complexity of the proposed IBBPSO and original algorithms and compare the differences.

6. Indicate whether uncertainty analysis or other forms of error analysis were performed

7. Discuss how the proposed method addresses partial observability and the curse of dimensionality. Elaborate on strategies employed to mitigate these issues.

8. Provide more comparisons between the proposed IBBPSO and other meta-heuristic algorithms. This will help confirm the superiority of the proposed method.

9. The paper lacks a robust evaluation of the proposed model. Benchmark your results against previous works to demonstrate the advantages and effectiveness of the methodology.

10. Clearly explain the limitations of this work and outline potential directions for future research.

6. PLOS authors have the option to publish the peer review history of their article (what does this mean? ). If published, this will include your full peer review and any attached files.

**Do you want your identity to be public for this peer review?** For information about this choice, including consent withdrawal, please see our Privacy Policy .

Reviewer #1: No

Reviewer #2: No

---

## [Author Response · Author response to Decision Letter 1]

3 Feb 2025

We would like to thank the reviewers and editor for their thoughtful comments and suggestions, which certainly help us to improve the quality of this paper. Specific as well as general changes are made according to the reviewers' comments, and are specifically highlighted in red in the revised manuscript. Below is a detailed explanation of the changes made to address all the concerns.For specific response content, please refer to the "Response to Reviewers" file.

---

## [Decision Letter · Decision Letter 1]

20 Feb 2025

Capacity optimization configuration and multi-dimensional value evaluation of integrated energy system with power-to-hydrogen

PONE-D-24-55307R1

Dear Dr. Sun,

We’re pleased to inform you that your manuscript has been judged scientifically suitable for publication and will be formally accepted for publication once it meets all outstanding technical requirements.

Kind regards,

Joy Nondy, Ph. D.

Academic Editor

PLOS ONE

Additional Editor Comments (optional):

Reviewers' comments:

Reviewer's Responses to Questions

**Comments to the Author**

1. If the authors have adequately addressed your comments raised in a previous round of review and you feel that this manuscript is now acceptable for publication, you may indicate that here to bypass the “Comments to the Author” section, enter your conflict of interest statement in the “Confidential to Editor” section, and submit your "Accept" recommendation.

Reviewer #1: All comments have been addressed

Reviewer #2: All comments have been addressed

2. Is the manuscript technically sound, and do the data support the conclusions?

Reviewer #1: Yes

Reviewer #2: Yes

3. Has the statistical analysis been performed appropriately and rigorously? 

Reviewer #1: Yes

Reviewer #2: Yes

4. Have the authors made all data underlying the findings in their manuscript fully available?

Reviewer #1: Yes

Reviewer #2: Yes

5. Is the manuscript presented in an intelligible fashion and written in standard English?

Reviewer #1: Yes

Reviewer #2: Yes

6. Review Comments to the Author

Reviewer #1: The authors have fully addressed all my concerns. Thanks for the revisions. No further comments for revisions.

Reviewer #2: (No Response)

7. PLOS authors have the option to publish the peer review history of their article (what does this mean? ). If published, this will include your full peer review and any attached files.

**Do you want your identity to be public for this peer review?** For information about this choice, including consent withdrawal, please see our Privacy Policy .

Reviewer #1: No

Reviewer #2: No

---

## [Editor Report · Acceptance letter]

PONE-D-24-55307R1

PLOS ONE

Dear Dr. Sun,

I'm pleased to inform you that your manuscript has been deemed suitable for publication in PLOS ONE. Congratulations! Your manuscript is now being handed over to our production team.

Kind regards,

on behalf of

Dr. Joy Nondy

Academic Editor

PLOS ONE